# OBJECT-CONSISTENT DISTILLATION FOR TEXT-TO-3D GENERATION

## ABSTRACT

Score Distillation Sampling (SDS) struggles to ensure that the pseudo ground truths from different viewpoints generated by the diffusion model correspond to the same 3D object in 3D generation. To analyze object inconsistency in SDS more directly and precisely, we theoretically model the renderings of a 3D object under continuous viewpoints as a connected subset of the image space. Based on this formulation, we introduce an object consistency constraint and identify two key sources of inconsistency: cross-view image discrepancy variation and cross-view distributional estimation error. In contrast to prior works, we focus on the former and propose Object-Consistent Distillation (OCD) which enforces the object consistency constraint during the generation of multi-view pseudo ground truths. Specifically, we estimate a dynamic object proxy using a sliding window and move the rendering of each viewpoint toward this proxy. We compare OCD with several recent generative baselines, and experiments demonstrate that OCD significantly mitigates irregular structures and unrelated artifacts in the generated objects. Code is provided in the supplemental material.

## 1 INTRODUCTION

In recent years, 3D generation has been attracting increasing attention due to its closer alignment with the physical world and its broad application prospects in fields such as human digitization [27], scene generation [3], and 3D editing [12]. However, training generative models typically demands large-scale datasets, and the collection of 3D data remains challenging due to factors such as acquisition difficulty and high annotation costs [38]. Fortunately, recent advancements in 2D generative models, especially diffusion models [15; 44; 35], have achieved remarkable success, reaching photo-realistic levels in terms of fidelity, diversity, and controllability. As a result, leveraging 2D generative models for 3D object synthesis holds great promise.

A prevailing paradigm for leveraging 2D generative models to synthesize 3D objects is Score Distillation Sampling (SDS) [33], whose core idea is to utilize diffusion models to supervise multi-view 2D renderings. Specifically, a 3D representation is rendered into 2D images from multiple camera viewpoints, which are then processed by a diffusion model to obtain denoised targets through a forward and reverse denoising process, serving as pseudo ground truth (pseudo-GT) for optimizing the 3D representation. Ideally, multi-view pseudo-GTs should correspond to consistent renderings of a single realistic 3D object. However, due to the one-to-many nature of the mapping between a text prompt and realistic images, the pseudo-GTs generated from different viewpoints may correspond to different 3D objects. As illustrated in Figure 1, varying only the rendering viewpoint while keeping the prompt fixed leads to pseudo-GTs that differ in shape, color, and background, a phenomenon we term object inconsistency. Such inconsistency can degrade the 3D optimization process, causing the resulting object to exhibit unnatural geometry or undesired artifacts. Recent works have investigated the inconsistency of multi-view images, and have proposed mitigating this issue by replacing the stochastic noise in SDS with more structured or deterministic noise [50; 23; 28]. While these methods have demonstrated improvements in visual coherence and clarity, they fall short of providing a comprehensive analysis of the root causes of object inconsistency. In this work, we first provide a theoretical perspective by characterizing the set of 2D renderings of a 3D object as a connected set in image space. Building on this insight, we show that the pseudo-GTs derived in the SDS paradigm may correspond to various underlying 3D objects due to the lack of object consistency constraints. Additionally, our analysis identifies two main sources of object inconsistency: cross-view image discrepancy variation and cross-view distributional estimation error. Unlike prior works [50; 23; 28]

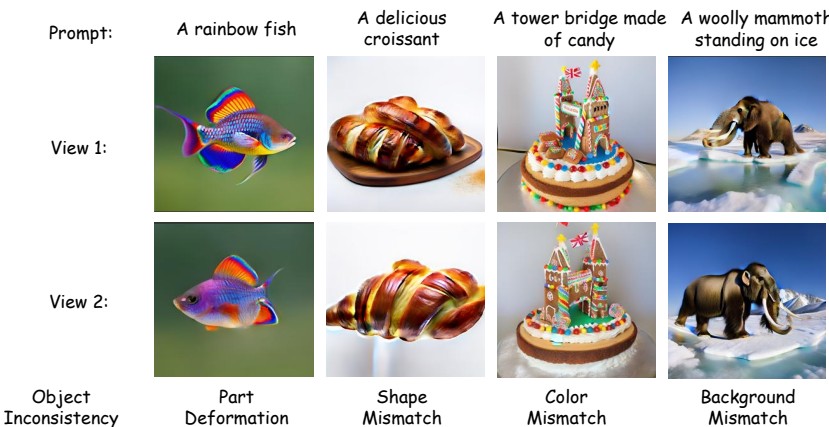

Figure 1: Given the same prompt, renderings from different viewpoints are input into the diffusion model, resulting in real images that correspond to inconsistent objects.

that primarily focus on the latter, we propose Object-Consistent Distillation (OCD), which explicitly enforces object consistency across multiple views. Specifically, we estimate a cross-view object proxy using previously sampled random viewpoints, serving as the representation of the target object, and each view-specific rendering is moved toward this estimated object proxy before being passed to the diffusion model. By introducing this object consistency constraint, the pseudo-GTs from different views are encouraged to align with the same underlying 3D object.

In summary, our main contributions are as follows:

- We conduct a theoretical analysis of the relationship between 3D objects and their 2D renderings, and formulate that under continuous viewpoint rendering, a 3D object corresponds to a connected subset of the image space.

- By examining the diffusion and denoising processes in diffusion models, we identify the sources of multi-view inconsistency in pseudo-GTs, attributing them to cross-view image discrepancy variation and cross-view distributional estimation error, noting that previous studies have largely neglected the former by focusing solely on the latter.

- We propose Object-Consistent Distillation (OCD), which moves view-specific renderings towards a cross-view object proxy, leading to more consistent and coherent supervision across views.

- We compare OCD with several state-of-the-art score distillation methods and demonstrate that it significantly improves generation fidelity and object coherence, while also alleviating the Janus problem, all with negligible additional computational and memory overhead.

## 2 RELATED WORK

**Diffusion Models.** The generation process of diffusion models consists of two key stages [40; 15; 41]: a forward diffusion process that gradually adds Gaussian noise to the image, and a reverse denoising process that reconstructs the image from the noisy input. By incorporating conditional prompts into the denoising process, the model can be guided to generate images that align with the given conditions [14]. Owing to their impressive generative capabilities, particularly in producing detailed and diverse samples, diffusion models have been widely adopted in various generative tasks, such as image generation [19; 25; 9], image super-resolution [34; 36], and image editing [32; 2] and are also considered promising for 3D generation [33; 45; 24; 26]. Despite the popularity of diffusion models for their diverse outputs [7; 4; 51], this diversity can compromise multi-view consistency when used as supervision for 3D reconstruction.

**Score Distillation Sampling for 3D Content Generation.** Score Distillation Sampling (SDS) [33; 45] is a paradigm that extends diffusion models, originally designed for 2D image generation, to the domain of 3D object synthesis. In this framework, a 3D representation is rendered into 2D images from multiple views, and a pretrained diffusion model provides supervision signals for these rendered images. Rather than requiring explicit 3D supervision, SDS leverages the generative power of large-scale diffusion models to indirectly guide the learning of 3D representations. Building

upon the successful text-to-3D paradigm established by SDS, subsequent studies have improved the generation process, achieving higher fidelity and greater diversity from multiple perspectives. For instance, they adopt annealed diffusion timestep strategies [18; 54; 22], introduce structured diffusion noise [28; 23; 50], decouple [6] or randomize 3D representations [47], and employ enhanced classifier-free guidance [52]. Nevertheless, the outputs of SDS often exhibit repeated patterns or structural collapse across views, which are manifestations of the multi-view consistency problem. To address this issue, mainstream approaches can be broadly divided into two categories. The first incorporates external prior knowledge, typically by leveraging additional datasets, 3D models or classifier [31; 39; 37; 5; 53], while the second enforces consistency regularizations within the original framework. The latter takes various forms, such as clipping the diffusion model's predicted scores [16], debiasing the input prompt [16; 1], introducing additional consistency losses [21; 48], aligning noise across different views [50; 23], and adopting adaptive view sampling strategies [17]. Unlike prior multi-view consistency methods, our approach analyzes the relationships across renderings, through which consistency constraints are effectively enforced.

## 3 PRELIMINARIES

**Diffusion Models.** Diffusion models are a class of deep generative models built upon two processes [15; 40]: a forward diffusion process that gradually adds Gaussian noise to real images, and a reverse denoising process that reconstructs the original images from pure Gaussian noise. Formally, given a data sample $x_0$ drawn from the real data distribution $p_0(x_0)$, the diffusion process incrementally adds Gaussian noise $\epsilon$ over time. At a specific time step $t$, the noisy data $x_t$ follows the distribution $\mathcal{N}(\sqrt{\bar{\alpha}_t}x_0, (1 - \bar{\alpha}_t)I)$, where $\bar{\alpha}_t = \prod_{i=1}^{t} \alpha_t$ and $\alpha_t$ is a time-dependent hyperparameter predefined by the diffusion schedule. In the reverse process, the diffusion model conditionally predicts the added noise at each time step given a prompt $y$. The model is then trained by minimizing the mean squared error (MSE) between the predicted noise $\epsilon_\phi(x_t, t, y)$ and the true noise $\epsilon$:

$$\mathcal{L}_{mse} = \mathbb{E}_{x_0, \epsilon, t}[\omega(t)||\epsilon_\phi(x_t, t, y) - \epsilon||_2^2], \tag{1}$$

where $\omega(t)$ denotes the weights at timestep $t$. After training, the denoised result can be obtained using the following simplified iterative formula:

$$x_{t-1} = \frac{1}{\sqrt{\alpha_t}}(x_t - \frac{1 - \alpha_t}{\sqrt{1 - \bar{\alpha}_t}}\epsilon_\phi(x_t, t, y)) \tag{2}$$

Through this formulation, pure noise is transformed into real data via the denoising process.

**Score Distillation Sampling.** Diffusion models have demonstrated remarkable performance in terms of both generation quality and diversity [7]. Leveraging this advantage, [33] first proposed the Score Distillation Sampling (SDS) framework, which utilizes the realistic images generated by a diffusion model as pseudo-GTs for 3D model optimization. Specifically, SDS first renders a randomly initialized 3D object representation $\theta$ into 2D images $g(\theta, c)$ from multiple viewpoints $c$. These images are perturbed with Gaussian noise at randomly sampled timestep $t$:

$$x_t(c) = \sqrt{\bar{\alpha}_t}g(\theta, c) + \sqrt{1 - \bar{\alpha}_t}\epsilon \tag{3}$$

Subsequently, the noisy images $x_t(c)$ are fed into the diffusion model $\phi$, which generates a conditional denoising prediction $\epsilon_\phi(x_t, t, y)$ guided by a given prompt $y$. Since SDS mirrors the two processes of diffusion models, the resulting gradient with respect to $\theta$ closely resembles Equation 1:

$$\nabla_\theta \mathcal{L}_{SDS} = \mathbb{E}_{t, \epsilon, c}[\omega(t)(\epsilon_\phi(x_t, t, y) - \epsilon)]\frac{\partial g(\theta, c)}{\partial \theta} \tag{4}$$

In the absence of direct 3D supervision, SDS successfully enables gradient updates to the 3D object by rendering it into images in a differentiable manner and supervising in the image space.

## 4 VARIABILITY OF UNDERLYING 3D OBJECTS IN SDS

By examining the gradient shown in Equation 4, it is observed that the generation of pseudo-GTs in SDS is performed independently for each viewpoint. Specifically, the diffusion model takes as input a rendering from a single viewpoint and performs diffusion and denoising operations under the guidance of a view-shared prompt. Ideally, each rendering carries view-specific information and, through the denoising process, reconstructs a realistic image of a realistic 3D object described by

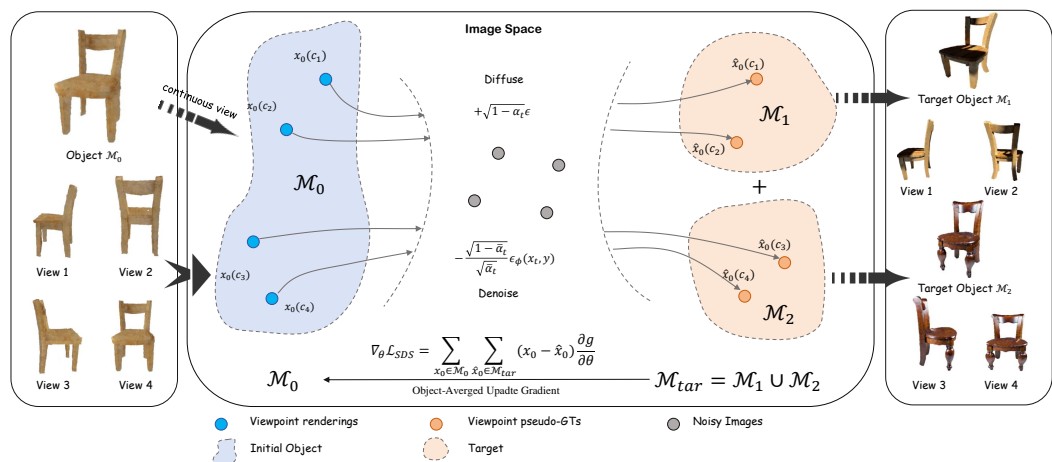

Figure 2: The relationship between 3D objects and their 2D renderings in SDS. The set of images obtained by rendering a 3D object from continuous viewpoints forms a connected subset of the image space. SDS randomly samples viewpoint images from this subset and applies independent noise and denoising to each. Since pseudo-GTs for each view are generated independently, the corresponding denoised pseudo-GTs are not guaranteed to be consistent. As a result, the optimization tends to drive the initial object towards an average of multiple potential target objects.

the prompt from its viewpoint. However, due to the inherently one-to-many relationship between a text prompt and realistic images, feeding renderings from different viewpoints as input can produce significantly different outputs, which may correspond to various underlying 3D objects. Since both the renderings and the pseudo ground truth are images, we analyze the variability of the underlying 3D objects in SDS within the image space.

First, the pseudo-GT $\hat{x}_0(c|y)$, obtained by feeding the rendered image under viewpoint $c$, $x_0(c) = g(\theta, c)$, into the diffusion model, is computed as:

$$\hat{x}_0(c|y) = \frac{x_t(c) - \sqrt{1 - \bar{\alpha}_t}\epsilon_\phi(x_t, t, y)}{\sqrt{\bar{\alpha}_t}} \tag{5}$$

Then, based on the diffusion process applied to $x_0(c)$, we can transform the noise alignment loss in Equation 4 into image alignment form:

$$\nabla_\theta \mathcal{L}_{SDS} = \mathbb{E}_{t,\epsilon,c}\left[\frac{\omega(t)}{\gamma(t)}(x_0(c) - \hat{x}_0(c|y))\right]\frac{\partial g}{\partial \theta} \tag{6}$$

Here, $\gamma(t) = \sqrt{1 - \bar{\alpha}_t}/\sqrt{\bar{\alpha}_t}$. Equation 6 reveals that the underlying principle of SDS is to supervise single-view images using pseudo-GTs generated by a diffusion model, thereby updating the 3D object representation. The central question is whether, by modifying only the input image to a diffusion model, the resulting outputs can consistently depict the same underlying object. Intuitively, this consistency is difficult to guarantee, as diffusion models are highly sensitive to input variations, and a single text prompt may correspond to a diverse set of real-world images. To examine this issue, we develop a theoretical framework aimed at representing 3D objects in image space, bridging the dimensional gap between images and objects. This enables a direct and intuitive representation of the 3D object indicated by the pseudo-GTs, thereby facilitating analysis of the variability of the underlying 3D objects.

We begin with an intuitive insight: for a 3D object, its rendering under a camera view $c$ corresponds to a single point in image space. As the view $c$ varies continuously, the corresponding renderings change continuously in image space. Motivated by this insight, we define the view parameter $c$ as a vector that encodes the coordinates determining the camera's position and orientation, as well as the location of a point light source [30], and proceed to state the following theorem.

**Theorem 1** (Connectedness of the Viewpoint–Image Set). *Let:*

- $C \subset \mathbb{R}^d$ *be a connected set representing the continuous viewpoint parameter space;*

- $\mathcal{I} = \mathbb{R}^N$ *be the image space, where $N = 3HW$ (flattening an $H \times W$ RGB image);*

- $g : C \to \mathcal{I}$ be a continuous rendering function, i.e., $g \in C^0(C, \mathcal{I})$.

*Then the image set $\mathcal{M} := g(C) \subset \mathcal{I}$ is connected.*

The formal proof of Theorem 1 is provided in the Appendix A.1. Based on Theorem 1, the optimization paradigm of SDS becomes more intuitive. As shown in Figure 2, within an initial object (the object to be optimized), denoted as an image set $\mathcal{M}_0$, SDS randomly samples a finite set of views and renders a set of images $\{x_0(c_1), x_0(c_2), \ldots, x_0(c_n)\}$. For each image, the diffusion model generates a corresponding pseudo-GT, $\{\hat{x}_0(c_1|y), \hat{x}_0(c_2|y), \ldots, \hat{x}_0(c_n|y)\}$. Next we analyze whether these pseudo-GTs could originate from the same object or, equivalently, from the same region.

If the pseudo-GTs generated from all sampled views correspond to the same target object $\mathcal{M}_{target}$, these randomly sampled viewpoint images are expected to lie in a connected set according to Theorem 1. To formalize the notion, we propose the following object consistency constraint:

$$||\hat{x}_0(c_1|y) - \hat{x}_0(c_2|y)||_2 \leq \delta(c1, c2), \tag{7}$$

where $\delta(c_1, c_2)$ denotes the distance between the pseudo-GTs of two viewpoints in the target object. While diffusion models are celebrated for their generative diversity [34; 15; 8], producing a wide range of realistic outputs depending on the input and injected noise, this very property becomes problematic in 3D generation tasks. When renderings originate from different viewpoints and are independently corrupted by noise, their resulting latent representations diverge significantly. As a consequence, the final outputs often fail to maintain consistency across views, violating the object consistency constraint in Equation 7. This ultimately results in inconsistencies in the underlying 3D object representations inferred from different viewpoints:

$$\nabla_\theta \mathcal{L}_{SDS} = \sum_{k=1}^{m} \mathbb{E}_{t,\epsilon} \left[ \frac{\omega(t)}{\gamma(t)} \sum_{x_0 \in \mathcal{M}_0} \sum_{\hat{x}_0 \in \mathcal{M}_{tar}} (x_0 - \hat{x}_0) \right] \frac{\partial g(\theta, c)}{\partial \theta}, \mathcal{M}_{tar} = \bigcup_{k=1}^{m} \mathcal{M}_k \tag{8}$$

Here, $\mathcal{M}_{tar}$ denotes the composite target object formed by the $m$ real objects to which the $n$ pseudo-labels. As a result, the initial object is implicitly encouraged to update toward all $m$ target objects simultaneously. In other words, the optimization process effectively treats the average of these $m$ objects as an object-level pseudo-GT, as illustrated in Figure 2. However, due to structural, color, and background variations among these target objects, this averaging process can introduce artifacts into the generated geometry, including unnatural structures or elements unrelated to the intended subject.

## 5 OBJECT-CONSISTENT SCORE DISTILLATION

To address the inconsistency of pseudo-GTs from different viewpoints with respect to the underlying 3D object, we investigate the generation process of viewpoint-specific pseudo-GTs, aiming to identify the contributing factors to such inconsistency. Specifically, for an image $x_0(c_i)$ rendered from an arbitrary viewpoint $c_i$, the pseudo-GT is obtained through a process of diffusion and denoising:

$$\hat{x}_0(c_i|y) = x_0(c_i) + \gamma(t_i)(\epsilon - \epsilon_\theta(x_{t_i}(c_i), t_i, y)), \quad i \in \{1, 2, \ldots, n\} \tag{9}$$

Since the combination of random noise $\epsilon$ and the predicted noise $\epsilon_\theta(x_{t_i}(c_i), t_i, y)$ are used to transform the original image into a realistic one, effectively estimating the discrepancy between the original and target distributions, we denote their combined noise as $\delta_D(c_i|y) = \gamma(t_i)(\epsilon - \epsilon_\theta(x_{t_i}(c_i), t_i, y))$. Subsequently, we replace the pseudo ground truth in the object consistency constraint in Equation 7:

$$||x_0(c_i) - x_0(c_j) + \delta_D(c_i|y) - \delta_D(c_j|y)||_2 \leq \delta(c_i, c_j), \quad \forall i \neq j, \; i, j \in \{1, 2, \ldots, n\} \tag{10}$$

It is observed that the failure of the object consistency constraint arises from two main sources: cross-view image discrepancy variation, $x_0(c_i) - x_0(c_j)$, and cross-view distributional estimation error, $\delta_D(c_i|y) - \delta_D(c_j|y)$. Most existing works [50; 23] attribute multi-view inconsistency solely to the latter, while overlooking the former. They argue that the randomness and independence of noise across different viewpoints prevent diffusion models from generating consistent pseudo ground truths. In contrast, we place greater emphasis on the former, which refers to the fact that discrepancies in the renderings themselves across different viewpoints can lead to variations in their positions in the latent space, thereby amplifying the diversity in the outputs of the diffusion model. For example, as shown in Figure 2, the inconsistencies between the 3D chairs corresponding to the pseudo-GTs of views 1 and 3 can be attributed approximately half to the diffusion and denoising process, and half to the intrinsic differences between the two renderings themselves.

Since our focus is on the differences between renderings from different viewpoints, we first decouple the distributional discrepancy from the cross-view image difference:

$$\|x_0(c_i) - x_0(c_j)\|_2 \leq \tilde{\delta}(c_i, c_j) \tag{11}$$

$$\tilde{\delta}(c_i, c_j) = \delta(c_i, c_j) - \|\delta_D(c_i) - \delta_D(c_j)\|_2 \tag{12}$$

In Equation 11, enforcing object consistency for a given viewpoint $c_i$ requires computing the distances between $c_i$ and all other sampled viewpoints, resulting in $n - 1$ pairwise constraints. To reduce the computational overhead, we propose to approximate this by using the distance between each viewpoint and an estimated object proxy of the target object, denoted as $O_i$:

$$\|x_0(c_i) - O_i\|_2 < \tilde{\delta}(c_i, O_i) \tag{13}$$

$$O_i = \sum_{j=i-l}^{j=i-1} \hat{x}_0(c_j) \tag{14}$$

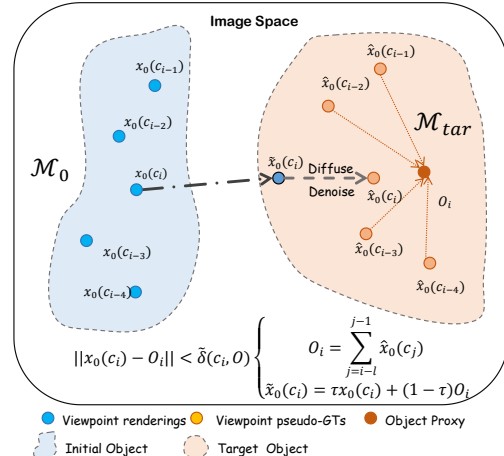

Figure 3: Overview of Object-Consistent Distillation. For each viewpoint, an object proxy is estimated based on the pseudo-GTs from preceding views. The current view's rendering is then moved toward this proxy, enforcing consistency such that the pseudo ground truths from different viewpoints correspond to the same target object.

Since the sampled viewpoints in SDS are optimized sequentially, we estimate the object proxy for each viewpoint using a sliding window. As shown in Equation 14 and Fig 3, the sliding window spans the preceding $l$ viewpoints relative to the current one, and the mean of the pseudo ground truths within the windowed views is taken as the object proxy. Note that the window size $l$ is small relative to the total number of views (e.g., about 1%), allowing the pseudo-ground-truths within the window to be approximated as renderings of the same target object.

Another issue lies in how to modify the input image of the diffusion model based on the object consistency constraint. Our intuition is that this constraint enforces the input image to be sufficiently close to the object proxy $O$, or equivalently, encourages the image to move towards the proxy. To this end, we adopt a moving average strategy to move the image towards the object proxy:

$$\tilde{x}_0(c_i) = \tau x_0(c_i) + (1 - \tau)O_i, \tag{15}$$

where we relax the distance $\tilde{\delta}$ to a momentum hyperparameter $\tau$. A smaller value of $\tau$ implies that the moved image is closer to the object proxy, which in turn corresponds to a tighter $\tilde{\delta}$ constraint.

By replacing the view image in Equatio 9 with the moved version, our method yields the update gradient for the 3D object representation as follows:

$$\nabla_\theta \mathcal{L}_{SDS} = \mathbb{E}_{t,\epsilon,c}[\frac{\omega(t)}{\gamma(t)}(x_0(c) - \tilde{x}_0(c) - \delta_D(c|y))]\frac{\partial g(\theta, c)}{\partial \theta} \tag{16}$$

It is important to emphasize that our method only modifies the images fed into the diffusion model to enforce consistency of target object, without altering the original per-view images themselves.

## 6 EXPERIMENTS

Following prior work [28; 50], we employ the same Stable Diffusion v2.1 [34] model and the Threestudio codebase [11]. We compare our proposed OCD method against three baselines: SDS [33], SDI [28], and CFD [50]. Specifically, these baselines differ in their approaches to estimating the distributional discrepancy $\delta_D$: SDS uses view-independent random noise, SDI leverages DDIM inversion to predict noise, and CFD employs cross-view consistent noise. Our final results combine OCD with these three baselines. Note that since the SDS code is not publicly available, we approximate SDS by replacing the cross-view consistent noise in CFD with view-independent random noise. Moreover, we adopt only the single-stage pipeline from CFD, i.e., directly distilling Stable Diffusion without any initialization. The momentum hyperparameter is set to $\tau = 0.9$ and , and the window length $l$ is defined as 1% of the total training steps (i.e., the total number of sampled views).Experiments are

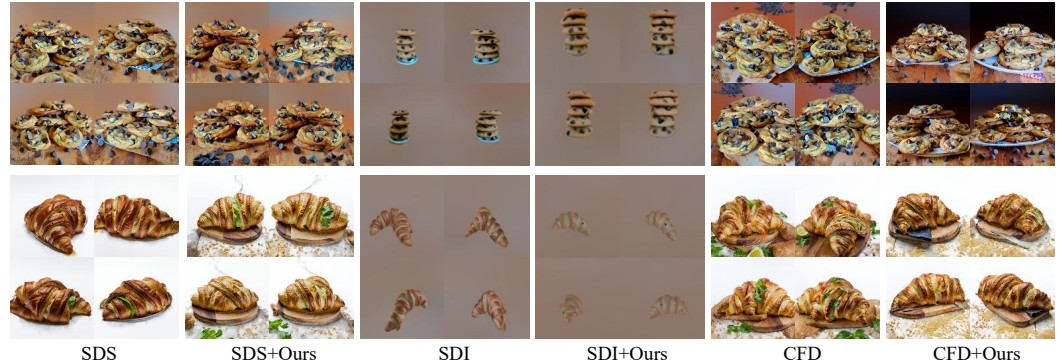

| SDS | SDS+Ours | SDI | SDI+Ours | CFD | CFD+Ours |

Figure 4: Comparison with other baselines on 3D generation. The first row uses the prompt "A plate piled high with chocolate chip cookies", while the second row uses "A delicious croissant".

Table 1: Quantitative comparisons to baselines for text-to-3D generation, evaluated by Eval3D [10] and CLIP IQA [46]. We report mean and standard deviation across 10 prompts and 50 views for each.

| Method | Eval3D (%) (↑) | | CLIP IQA (%) | | | IR (↑) | Time | VRAM |
|---|---|---|---|---|---|---|---|---|
| | Geometric | Structural | "natural"↑ | "real"↑ | "complexity"↓ | | | |
| SDS [33] | $56.66 \pm 25.34$ | $85.92 \pm 5.24$ | $39 \pm 9.6$ | $40 \pm 6.1$ | $74 \pm 7.7$ | $-0.47 \pm 0.18$ | 150min | 14.4GB |
| SDS+ours | $\mathbf{60.51 \pm 24.55}$ | $\mathbf{87.23 \pm 1.68}$ | $\mathbf{47 \pm 8.4}$ | $\mathbf{67 \pm 10.1}$ | $\mathbf{69 \pm 8.1}$ | $\mathbf{0.23 \pm 0.10}$ | 151min | 14.4GB |
| SDI [28] | $76.39 \pm 19.9$ | $85.15 \pm 4.11$ | $18 \pm 6.6$ | $7 \pm 3.8$ | $63 \pm 8.3$ | $-2.10 \pm 0.19$ | 83min | 7.9GB |
| SDI+Ours | $\mathbf{89.36 \pm 2.51}$ | $\mathbf{87.11 \pm 1.55}$ | $\mathbf{55 \pm 5.2}$ | $\mathbf{21 \pm 6.8}$ | $\mathbf{53 \pm 6.7}$ | $\mathbf{-2.03 \pm 0.17}$ | 88min | 7.9GB |
| CFD [50] | $66.77 \pm 19.69$ | $86.45 \pm 1.41$ | $42 \pm 5.9$ | $60 \pm 7.0$ | $73 \pm 4.3$ | $-0.14 \pm 0.15$ | 152min | 14.4GB |
| CFD+ours | $\mathbf{70.10 \pm 23.21}$ | $\mathbf{88.46 \pm 1.61}$ | $\mathbf{52 \pm 4.4}$ | $\mathbf{71 \pm 5.6}$ | $\mathbf{70 \pm 6.3}$ | $\mathbf{0.26 \pm 0.15}$ | 155min | 14.4GB |

conducted on RTX 3090 GPUs, and images are generated at a resolution of $128 \times 128$. More detailed experimental settings and higher-resolution results can be found in Appendix B, C, and D.

**Qualitative Comparision** Figure 4 presents a comparison between our method and other approaches that adopt different noise configurations. In the generation results of SDS [33] with random noise and CFD [50] with multi-view consistent noise, the main objects often appear cluttered, and unrelated content may emerge in the background. This is because the pseudo-GTs derived from independent views correspond to different underlying objects. Under the same prompt, such as "cookies", the placement, shape, and background context of the cookies vary significantly across different viewpoints. As a result, the inconsistent pseudo-GTs lead to outputs that blend multiple possible objects. Moreover, both SDI [28] and CFD sometimes produce objects with distorted or unrealistic structures. This issue also stems from the inconsistency among the 3D objects implied by the pseudo-GTs across views. For example, in the croissant generated by SDI, since the pseudo-GTs corresponding to the front and side views conflict, the resulting geometry becomes a malformed artificial shape. In contrast, our method incorporates an object-level consistency constraint, which significantly improves the coherence of the generated results. By moving the images away from regions that are likely to cause conflicts in object shape or position across different viewpoints, our approach reduces inconsistency and aligns the generation toward a more unified 3D representation.

**Quantitive Comparision** Following prior works [28; 33; 52], we adopt CLIP [46] and ImageReward (IR) [49], and additionally incorporate Eval3D [13] to quantitatively assess the quality of the generated results, aiming to approximate human perception of the synthesized objects. We first adopt the "Geometric Consistency" and "Structural Consistency" metrics from Eval3D [13] to evaluate the multi-view consistency of the generated objects. As shown in Table 1, by introducing object proxies to impose consistency constraints, geometric consistency and structural consistency are improved by 6.72% and 1.76% on average, respectively. Next, we include the CLIP Image Quality Assessment (IQA) [46] to evaluate the naturalness, realism, and complexity of the rendered views, achieving an average improvement of 13.89%. It is worth noting that the first two metrics measure the realism of the renderings, while the last reflects the presence of irrelevant artifacts. On the IR metric, which is designed to mimic human preferences, OCD significantly boosts the performance of both SDS and CFD methods, and yields a slight improvement for the SDI method, with an average gain of 0.39. This demonstrates the advantage of OCD in enhancing the realism of the generated 3D objects. It is indicated that perturbing the input images effectively encourages the output images to converge toward a consistent set of target objects, thereby producing 3D reconstructions that are more

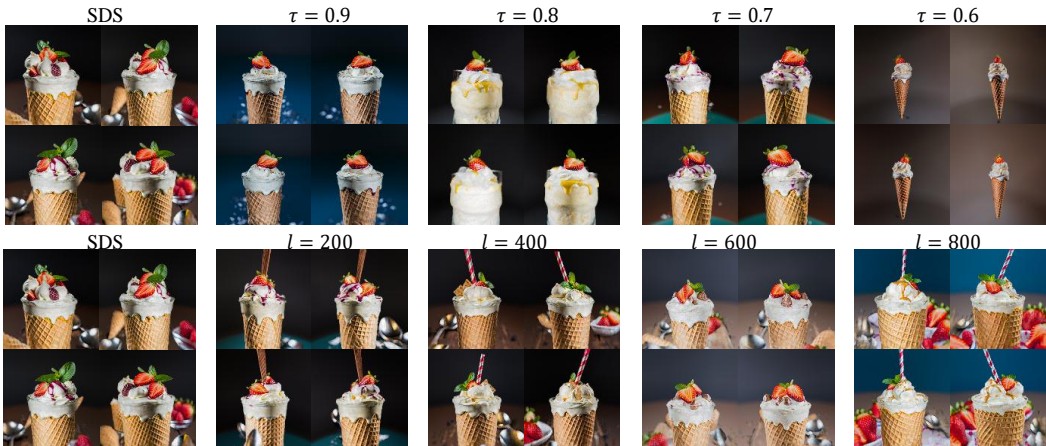

Figure 5: Effect of momentum hyperparameter $\tau$ and window size $l$ on generation. Prompt: "A DSLR photo of an ice cream sundae".

faithful to real-world objects. Finally, the last two columns of Table 1 present the training time and memory usage of different methods. As shown, our method incurs less than 1% additional overhead, highlighting its computational efficiency and demonstrating its advantage as a resource-efficient solution for enforcing object-level consistency.

## 6.1 ABLATION STUDIES

**Ablation on momentum hyperparameter $\tau$.** The first row of Figure 5 shows the effect of varying the momentum hyperparameter $\tau$ on the generation quality. As $\tau$ decreases, the generated results become increasingly focused on the target object described by the prompt, while substantially reducing irrelevant elements introduced by the prompt's ambiguous semantics. This is because a smaller $\tau$ places greater emphasis on the estimated object proxy, which encapsulates semantics shared across multiple views, thereby substantially reducing the semantic ambiguity within each individual view.

**Ablation on window size $l$.** The second row of Figure 5 illustrates the effect of varying the window size on the performance of our method. As the window size increases, the generated results exhibit undesired and irrelevant artifacts, and the overall appearance becomes increasingly similar to that of SDS. This suggests that when the window size is large, the composite target object in Equation 8 encompasses a larger set of real objects. As a result, the estimated object proxy tends to approximate an average over multiple objects, which severely compromises the object consistency among pseudo-GTs. In contrast, when a smaller window size is used, the variability of the underlying 3D objects is substantially reduced, making the generations more faithful to the prompt.

## 6.2 GENERATION DIVERSITY

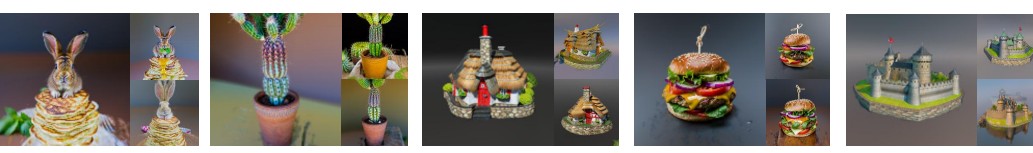

Figure 6: 3D generated objects under different random seeds. The prompts from left to right are: *A baby bunny sitting on top of a stack of pancakes*; *A small saguaro cactus planted in a clay pot*; *A 3D model of an adorable cottage with a thatched roof*; *A hamburger*; *A highly detailed DSLR photo of a 3D model of a historical stone castle*.

A potential concern is that aligning the pseudo-GTs from all viewpoints toward a object proxy might compromise the generation diversity. Figure 6 shows results under different seeds, demonstrating that our method preserves diversity while improving consistency. This benefit stems from the momentum-based moving average used to estimate the proxy, which balances current view information with a stable reference, avoiding over-constraining the diffusion process.

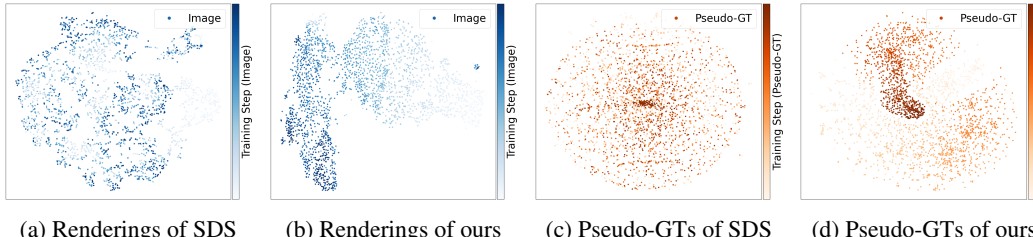

| (a) Renderings of SDS | (b) Renderings of ours | (c) Pseudo-GTs of SDS | (d) Pseudo-GTs of ours |

Figure 7: T-SNE visualization of renderings and pseudo-GTs in the image space. (a)/(c): Distributions of viewpoint renderings and pseudo-GTs obtained with SDS. (b)/(d): Corresponding distributions obtained with OCD. Darker colors indicate samples from later training steps.

### 6.3 IMAGE SPACE VISUALIZATION

To more directly and clearly demonstrate the effect of introducing the object proxy on 3D object generation, we visualize the renderings and pseudo-GTs during the optimization process using t-SNE [43]. As shown in Figure 7 (a) and (c), before applying the object consistency constraint, the distributions of renderings and pseudo-ground-truths remain dispersed and show little change throughout optimization. In contrast, after introducing the constraint, as shown in (b) and (d), both renderings and pseudo-GTs progressively concentrate and converge toward a consistent single target object. This difference arises from the object proxy, which conditions each pseudo-GT on previous views during optimization, resulting in a more concentrated distribution. Consequently, the supervisory signals enforce tighter consistency in 3D renderings from different views.

### 6.4 JANUS PROBLEM

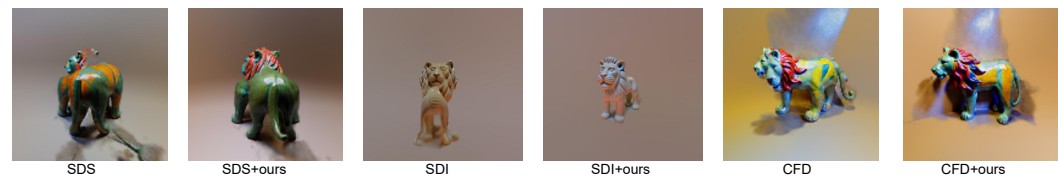

| SDS | SDS+ours | SDI | SDI+ours | CFD | CFD+ours |

Figure 8: Effect of OCD on the Janus Problem. Prompt: a ceramic lion.

The Janus problem refers to the phenomenon where a generated 3D object inaccurately presents the canonical view from multiple viewpoints. This typically arises when the pseudo-GTs generated by the diffusion model for a given view does not match that view, for instance, producing a front-facing image when conditioned on a back view rendering. Since OCD is designed to enhance multi-view consistency, it proves effective in alleviating this problem. n Figure 8 and Table 1, we qualitatively and quantitatively demonstrate the effectiveness of OCD in alleviating the Janus problem. these baselines either do not impose any multi-view consistency constraints or only enforce consistency from the perspective of noise, yet all exhibit severe manifestations of the Janus problem. In contrast, applying OCD significantly alleviates this issue. This highlights the critical role of the renderings' positions in the latent space in influencing the diffusion model's output. Through the utilization of cross-view information, our method provides more consistent multi-view supervisory signals, thereby enhancing the realism and coherence of the generated 3D objects.

## 7 CONCLUSION

By modeling the rendering of a 3D object under continuous viewpoints as a connected subset of the image space, we provide a more intuitive and effective formulation for the Score Distillation Sampling (SDS) paradigm. To analyze the variability of 3D objects corresponding to the pseudo-GTs produced by diffusion models, we introduce an object consistency constraint. Integrating this constraint into the pseudo-GT generation process allows us to attribute multi-view inconsistency to cross-view image discrepancy variation and cross-view distributional estimation error. Focusing on the inconsistencies caused by the former, we propose Object-Consistent Distillation (OCD), which incorporates object-consistency constraints during the generation of multi-view pseudo ground truths. Specifically, a view-dependent sliding window is used to estimate an object proxy, and renderings from each viewpoint are moved toward this proxy before being fed into the diffusion model. Experimental results show that OCD enhances generation fidelity and object coherence, while also contributes to alleviate the Janus problem with minimal overhead.

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

# A  Proof

## A.1  Proof the Theorem 1

In this subsection, we present the proof of Theorem 1: Connectedness of the Viewpoint–Image Set. We begin by discussing the rationale behind the assumptions of the theorem.

Assumption 1: $C \subset \mathbb{R}^d$ be a connected set representing the continuous viewpoint parameter space. We define the viewpoint parameter $c \in C \subset \mathbb{R}^d$ as a vector that encodes both the *camera configuration* and the *position of a point light source*. Specifically,

$$c = [\mu_{\text{cam}}, \zeta_{\text{cam}}, dis_{\text{cam}}, la_{\text{cam}}, up_{\text{cam}}, x_{\text{lig}}, y_{\text{lig}}, z_{\text{lig}}]$$

where:

- $\mu_{\text{cam}}, \zeta_{\text{cam}}, dis_{\text{cam}}$ denote the *elevation angle*, *azimuth angle*, and *distance from the origin*, respectively. Together, they determine the camera position in spherical coordinates.

- $la_{\text{cam}}$ and $up_{\text{cam}}$ represent the camera's *look-at* direction and *up* vector, which define its orientation.

- $[x_{\text{lig}}, y_{\text{lig}}, z_{\text{lig}}]$ specifies the location of a *point light source*.

All of these components are continuous variables. For instance, we define

$$\mu_{\text{cam}} \in [-90°, 90°], \quad \zeta_{\text{cam}} \in [0°, 360°], \quad dis_{\text{cam}} \in [0, 2],$$

$$la_{\text{cam}} \in \mathbb{S}^2 = \{\mathbf{x} \in \mathbb{R}^3 \mid \|\mathbf{x}\| = 1\},$$

$$up_{\text{cam}} \in \mathbb{S}^2 = \{\mathbf{x} \in \mathbb{R}^3 \mid \|\mathbf{x}\| = 1\},$$

$$x_{\text{lig}}, y_{\text{lig}}, z_{\text{lig}} \in \mathbb{R}.$$

Since each component varies continuously in $\mathbb{R}$, the resulting parameter space $C$ is a connected subset of $\mathbb{R}^d$.

Assumption 2: $\mathcal{I} = \mathbb{R}^N$ be the image space, where $N = 3HW$ (flattening an $H \times W$ RGB image). The image space $\mathcal{I}$ is often taken as $\mathbb{R}^N$ with $N = 3HW$ by flattening an $H \times W$ RGB image. Since each pixel channel value is normalized and constrained within the continuous interval $[0, 1]$, the realistic image space is actually a bounded, continuous subset: $\mathcal{I} \subseteq [0, 1]^N \subset \mathbb{R}^N$. This subset forms a compact and connected space under the standard Euclidean topology, ensuring that the image representations vary continuously with respect to pixel intensity changes.

Assumption 3: $g : C \to \mathcal{I}$ be a continuous rendering function, i.e., $g \in C^0(C, \mathcal{I})$. In the SDS rendering pipeline [33], the viewing parameter uniquely determines the camera's projection center. From this center, rays are cast through each pixel into the scene using only continuous operations such as matrix multiplication and vector addition. Thus, the mapping from viewing parameters to ray directions is continuous. Subsequently, SDS samples points densely along each ray in 3D space. Although this sampling is discrete, the high sampling density allows us to approximate the process as a continuous integral along the ray. Specifically, the sampled points $\{\mathbf{p}(s_i)\}$ represent discrete locations along the ray, where each point is defined as

$$\mathbf{p}(s_i) = \mathbf{o} + s_i\mathbf{d},$$

with $\mathbf{o}$ being the camera origin and $\mathbf{d}$ the ray direction. Here, $s_i$ denotes the distance from the camera origin along the ray.

At each sampled point $\mathbf{p}(s_i)$, the multi-layer perceptron (MLP) is queried to produce the volumetric density $\rho(\mathbf{p}(s_i))$ and the view-dependent RGB color $\mathbf{cl}(\mathbf{p}(s_i))$. Although the sampling is discrete, the dense sampling allows the discrete summation of these values to closely approximate the continuous volume rendering integral:

$$C(\mathbf{r}) = \int_{s_{\min}}^{s_{\max}} S(s)\,\rho(\mathbf{p}(s))\,\mathbf{cl}(\mathbf{p}(s))\,ds,$$

$$S(s) = \exp\left(-\int_{s_{\min}}^{s} \rho(\mathbf{p}(u))\,du\right),$$

where $C(\mathbf{r})$ is the final pixel color along ray $\mathbf{r}$, $S(s)$ is the accumulated transmittance. The process of mapping a view parameter $\mathbf{c}$ to the RGB color of a pixel can be viewed as a composition of several continuous functions. First, the view parameter $\mathbf{c}$ determines the camera origin $\mathbf{o}$ and direction $\mathbf{d}$ in a continuous manner. Then, for each depth value $s \in [s_{\min}, s_{\max}]$, the point along the ray is given by $\mathbf{p}(s) = \mathbf{o} + s\mathbf{d}$, which is also continuous in $\mathbf{o}$ and $\mathbf{d}$, hence continuous in $\mathbf{cl}$. The volumetric density $\rho(\cdot)$ and color $\mathbf{cl}(\cdot)$ are computed by a multi-layer perceptron (MLP), which is a composition of continuous functions and thus itself continuous. Finally, the volume rendering integral is continuous with respect to $\mathbf{o}$ and $\mathbf{d}$, and hence with respect to $\mathbf{c}$. Therefore, the overall rendering function $g : C \mapsto \mathcal{I}$ can be regarded as a continuous function.

Given the assumptions of Theorem 1, $\mathcal{M} := g(C)$ is connected in $\mathcal{I}$ by the basic topological fact that continuous maps preserve connectedness [42].

## A.2 Deformation of the gradient formulation in SDS

In the main paper, we transform the noise-alignment gradient formulation of SDS (Equation 4 in the main text), into the image-alignment gradient formulation (Equation 6 in the main text). Below, we provide the detailed derivation of this transformation.

$$\nabla_\theta \mathcal{L}_{SDS} = \mathbb{E}_{t,\epsilon,c}[\omega(t)(\epsilon_\phi(x_t,t,y) - \epsilon)]\frac{\partial g(\theta,c)}{\partial \theta} \tag{17}$$

$$= \mathbb{E}_{t,\epsilon,c}[\omega(t)(\epsilon_\phi(x_t,t,y) - \epsilon) + \omega(t)\frac{(x_t(c) - x_t(c))}{\sqrt{1-\bar{\alpha}_t}}]\frac{\partial g(\theta,c)}{\partial \theta} \tag{18}$$

$$= \mathbb{E}_{t,\epsilon,c}[\omega(t)\frac{x_t(c) - \sqrt{1-\bar{\alpha}_t}\epsilon}{\sqrt{1-\bar{\alpha}_t}} - \omega(t)\frac{x_t(c) - \sqrt{1-\bar{\alpha}_t}\epsilon_\phi(x_t,t,y)}{\sqrt{1-\bar{\alpha}_t}}]\frac{\partial g(\theta,c)}{\partial \theta} \tag{19}$$

$$= \mathbb{E}_{t,\epsilon,c}[\frac{\omega(t)}{\gamma(t)}\frac{x_t(c) - \sqrt{1-\bar{\alpha}_t}\epsilon}{\sqrt{\bar{\alpha}_t}} - \frac{\omega(t)}{\gamma(t)}\frac{x_t(c) - \sqrt{1-\bar{\alpha}_t}\epsilon_\phi(x_t,t,y)}{\sqrt{\bar{\alpha}_t}}]\frac{\partial g(\theta,c)}{\partial \theta} \tag{20}$$

$$= \mathbb{E}_{t,\epsilon,c}[\frac{\omega(t)}{\gamma(t)}(x_0(c) - \hat{x}_0(c|y))]\frac{\partial g(\theta,c)}{\partial \theta} \tag{21}$$

$$\tag{22}$$

# B Implementation details

## B.1 Experiment Configuration

Our experiments are conducted on 8 NVIDIA RTX 3090 GPUs, each equipped with 24GB of VRAM. Since the implementation of the SDI [28] baseline requires larger GPU memory, we adopt a resolution of $128 \times 128$ in the experiments involving SDI, specifically in parts of Figures 4 and 8 in the main text and Figure 11 in the Appendix, to avoid memory overflow. Nevertheless, these results still demonstrate the effectiveness of our method. We also present the results of applying OCD on top of the CFD [50] and SDS [33] baselines at a resolution of $512 \times 512$ in Figures 10, 12, 13 and 14, which showcase finer details and high-fidelity generation quality. For the teacher diffusion model, we follow most prior works[23; 28; 50] and adopt Stable Diffusion v2.1 [34]. It is worth noting that Stable Diffusion v2.1, as a Latent Diffusion Model, performs the noising and denoising processes in the latent space rather than in the pixel space. Accordingly, the object proxy computations are also carried out in the latent space of the diffusion model. Additionally, several studies have shown that timestep annealing [28; 18; 54] and the use of extra negative prompts [20; 29; 50] are beneficial for 3D generation; thus, we incorporate both strategies in our framework. By default, we use the same classifier-free guidance (CFG) scale of 7.5. Except for applying our method to SDI, where 10k training steps and a learning rate of $10^{-2}$ are used, we adopt 25k steps and a learning rate of $10^{-3}$ in all other cases.

## B.2 Algorithm

Algorithm 1 and 2 illustrate the difference between the original SDS pipeline and our proposed OCD method. The key distinction lies in that OCD incorporates cross-view information before feeding the image into the diffusion model. The differences between OCD and SDS are highlighted in red.

---

**Algorithm 1** Dreamfusion (SDS)

---

**Require:** $\theta$ - 3D object representation
  $c$ - camera view parameter
  $y$ - text prompt
  $g : \mathcal{C} \to \mathcal{I}$ - differentiable renderer
  $\epsilon_\phi^{(t)}$ - trained diffusion model
**Ensure:** 3D shape $\theta$ of $y$
  **procedure** DREAMFUSION($y$)
    **for** $i$ in range($n\_iters$) **do**
      $t \leftarrow \text{Uniform}(0, 1)$
      $c \leftarrow \text{Uniform}(\mathcal{C})$
      $\epsilon \leftarrow \text{Normal}(0, I)$
      $x_t \leftarrow \sqrt{\bar{\alpha}(t)} g(\theta, c) + \sqrt{1 - \bar{\alpha}(t)} \epsilon$
      $\nabla_\theta \mathcal{L}_{SDS} = \sigma(t) \left[ \epsilon_\phi^{(t)}(x_t, y) - \epsilon \right] \frac{\partial g}{\partial \theta}$
      Backpropagate $\nabla_\theta \mathcal{L}_{SDS}$
      SGD update on $\theta$

---

**Algorithm 2** Ours (OCD)

---

**Require:** $\theta$ - 3D object representation
  $c$ - camera view parameter
  $y$ - text prompt
  $g : \mathcal{C} \to \mathcal{I}$ - differentiable renderer
  $\epsilon_\phi^{(t)}$ - trained diffusion model
  $\tau$ - strength of the object consistency constraint
**Ensure:** 3D shape $\theta$ of $y$
  **procedure** OURS($y$)
    **for** $i$ in range($n\_iters$) **do**
      $t \leftarrow Time\_annealing(i)$
      $c \leftarrow \text{Uniform}(\mathcal{C})$
      $O_i \leftarrow \sum_{j-l}^{j=i-1} \hat{x}_0(c_j)$
      $\tilde{x} \leftarrow \tau g(\theta, c) + (1 - \tau) O_i$
      $\epsilon \leftarrow \text{Normal}(0, I)$
      $x_t \leftarrow \sqrt{\bar{\alpha}(t)} \tilde{x} + \sqrt{1 - \bar{\alpha}(t)} \epsilon$
      $\nabla_\theta \mathcal{L}_{SDS} = \sigma(t) \left[ \epsilon_\phi^{(t)}(x_t, y) - \epsilon \right] \frac{\partial g}{\partial \theta}$
      $\hat{x}_0(c_i) \leftarrow g(\theta, c) + \gamma(t)(\epsilon - \epsilon_\theta(x_t(c), t, y))$
      Backpropagate $\nabla_\theta \mathcal{L}_{SDS}$
      SGD update on $\theta$

---

In Section 5, we identify two primary sources of object inconsistency: cross-view image discrepancy variation and cross-view distributional estimation error. Since prior work has largely overlooked the former, we proposed OCD to explicitly address cross-view image discrepancies. In Section 6, we compare the performance of OCD combined with various cross-view distributional estimation methods. In this section, we provide additional results using various distributional estimation techniques in conjunction with OCD, and further analyze how the combination of these two factors influences the quality of the generated 3D content. In Figure 9, we demonstrate the combination of OCD with the original SDS [33]; in Figure 10, we show the results of combining OCD with CFD [50]; and in Figure 11, we present the results of combining OCD with SDI [28]. All results consistently demonstrate that the introduction of OCD significantly improves the original generation quality, effectively eliminating implausible structures or components, and even leading to a noticeable reduction of the Janus problem.

## C ADDITIONAL GENERATION

In Figure 12, we present additional high-fidelity 3D object generation results obtained using the OCD algorithm illustrated in Algorithm 2, including examples with multiple objects and complex prompts. OCD consistently achieves high-fidelity and realistic results across diverse generation

Table 2: Quantitative comparisons to baselines for text-to-3D generation, evaluated by Eval3D [10] and CLIP IQA [46]. We report mean and standard deviation across 10 prompts and 50 views for each.

| Method | Eval3D (%) (↑) | | CLIP IQA (%) | | | IR (↑) | CLIP Score |
|---|---|---|---|---|---|---|---|
| | Geometric | Structural | "natural"↑ | "real"↑ | "complexity"↓ | | |
| SDS [33] | $56.66 \pm 25.34$ | $85.92 \pm 5.24$ | $39 \pm 9.6$ | $40 \pm 6.1$ | $74 \pm 7.7$ | $-0.47 \pm 0.18$ | $22.87 \pm 0.05$ |
| SDS+ours | $\mathbf{60.51 \pm 24.55}$ | $\mathbf{87.23 \pm 1.68}$ | $\mathbf{47 \pm 8.4}$ | $\mathbf{67 \pm 10.1}$ | $\mathbf{69 \pm 8.1}$ | $\mathbf{0.23 \pm 0.10}$ | $\mathbf{23.15 \pm 0.03}$ |
| SDI [28] | $76.39 \pm 19.9$ | $85.15 \pm 4.11$ | $18 \pm 6.6$ | $7 \pm 3.8$ | $63 \pm 8.3$ | $-2.10 \pm 0.19$ | $21.16 \pm 0.03$ |
| SDI+Ours | $\mathbf{89.36 \pm 2.51}$ | $\mathbf{87.11 \pm 1.55}$ | $\mathbf{55 \pm 5.2}$ | $\mathbf{21 \pm 6.8}$ | $\mathbf{53 \pm 6.7}$ | $\mathbf{-2.03 \pm 0.17}$ | $\mathbf{21.26 \pm 0.03}$ |
| CFD [50] | $66.77 \pm 19.69$ | $86.45 \pm 1.41$ | $42 \pm 5.9$ | $60 \pm 7.0$ | $73 \pm 4.3$ | $-0.14 \pm 0.15$ | $22.89 \pm 0.02$ |
| CFD+ours | $\mathbf{70.10 \pm 23.21}$ | $\mathbf{88.46 \pm 1.61}$ | $\mathbf{52 \pm 4.4}$ | $\mathbf{71 \pm 5.6}$ | $\mathbf{70 \pm 6.3}$ | $\mathbf{0.26 \pm 0.15}$ | $\mathbf{22.90 \pm 0.08}$ |

scenarios, covering single and multiple objects, simple and complex structures, and scenes with complex backgrounds. More importantly, the renderings from different viewpoints consistently correspond to the same underlying real-world object.

## D    ADDITIONAL COMPARISION

Following prior works [28; 33; 52], we additionally report CLIP Score in Table 2 for quantitative comparison. This metric provides an overall evaluation of the generation quality across different methods. Since our approach primarily focuses on producing natural and realistic object geometry with strong multi-view consistency, the improvement on the CLIP Score metric is limited, though it still remains at a competitive level.

In Section 2, we point out that numerous strong approaches for text-to-3D generation have been proposed. To highlight the advantages of our method, we compare it against several representative baselines in the main text and below. In Figures 13 and 14, we present 3D objects generated at a resolution of $512 \times 512$ using the algorithm described in the Algorithm 2, and compare them against several baselines, including DreamFusion [33], Magic3D [24], NFSD [20], SDI [28], ISM [23], HiFA [54], Fantasia3D [6] and CFD [50]. It is worth noting that some comparisons are limited due to the unreproducibility of certain methods [17; 23], while others rely on additional external information [31; 39; 37; 5; 53], which would lead to an unfair comparison. The results demonstrate that OCD significantly improves the realism of generated 3D objects while preserving fine-grained details at high resolutions. This enhancement is evident in multiple aspects. First, the generated objects exhibit a notable reduction in unnatural artifacts or ambiguous structures that are often present in baseline methods. Second, the geometric configuration of the objects becomes more coherent and semantically meaningful, with clearer contours and physically plausible shapes. These findings suggest that the integration of OCD contributes not only to higher visual fidelity but also to better alignment with the real-world physical characteristics of the target objects.

## E    LIMITATIONS

While our proposed OCD method demonstrates strong performance in enhancing cross-view consistency and alleviating the Janus problem, it is not without limitations. First, the introduction of an object-level proxy into the generation process may potentially introduce bias into the generations. This proxy acts as an intermediate representation across views, which could influence the generated content in unintended ways. We have not yet conducted a systematic study on how such bias may manifest or how it might impact different object categories, viewpoints, or styles. Second, although OCD shows promising results in mitigating the Janus problem, suggesting a strong connection between object-level consistency and view-dependent artifacts, we have not thoroughly explored the theoretical or empirical relationship between the two. In particular, it remains unclear whether enforcing cross-view consistency alone is sufficient to fully eliminate the problem, or whether additional geometric or semantic constraints are necessary. Finally, In our current implementation, the object-consistency constraint strength, denoted as $\tau$, is uniformly applied across all camera viewpoints. While this simplification enables stable optimization and reduces hyperparameter tuning complexity, it may limit the expressiveness and adaptability of the method. A uniform consistency strength may under-constrain some views while over-constraining others, potentially impeding the generation of more view-consistent or multi-object genreation.

SDS                                    Ours

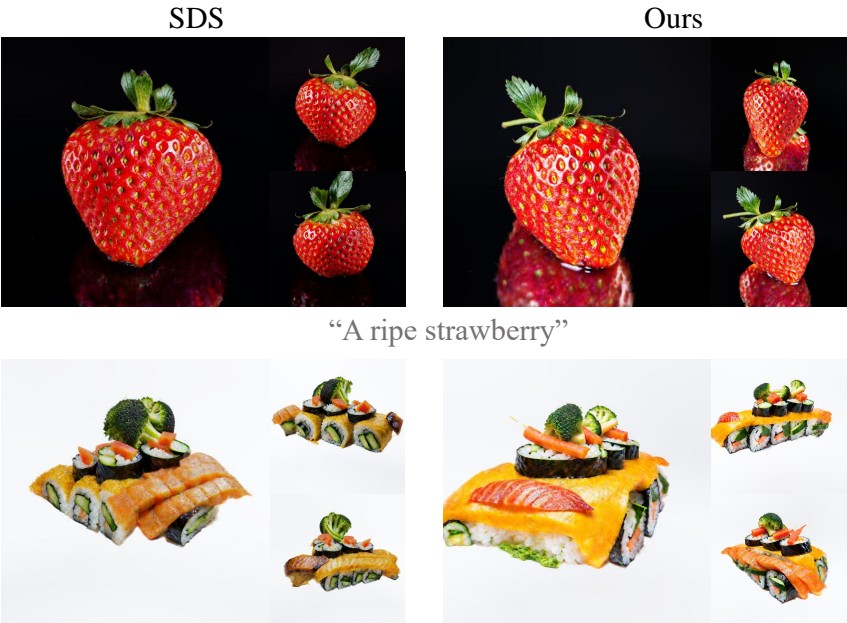

"A ripe strawberry"

"A car made of sushi"

Figure 9: Comparison between SDS and SDS combined with OCD (Ours).

CFD

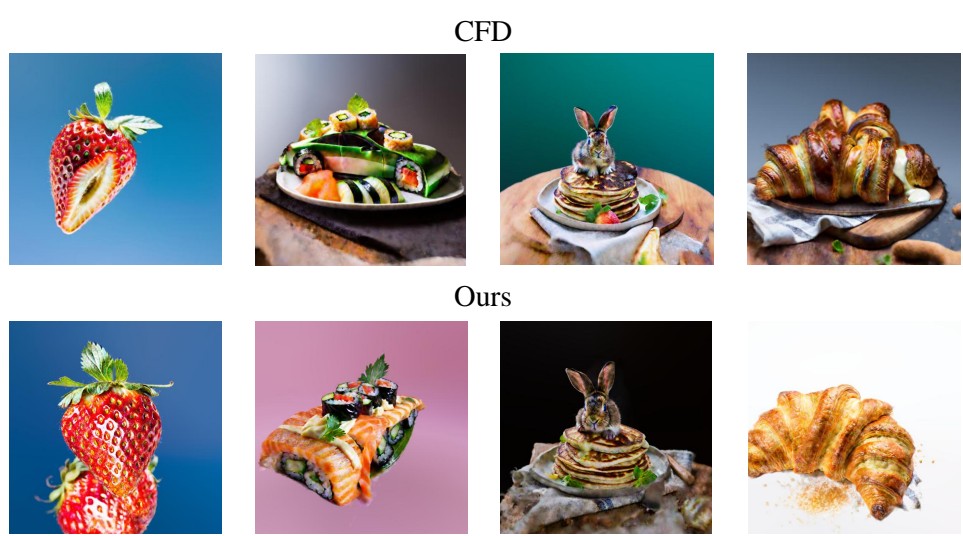

Ours

Figure 10: Comparison between CFD and CFD combined with OCD (Ours). From left to right, the prompts used are: "a ripe strawberry," "a car made of sushi," "a baby bunny sitting on top of a stack of pancakes," and "a delicious croissant."

SDI                                          Ours

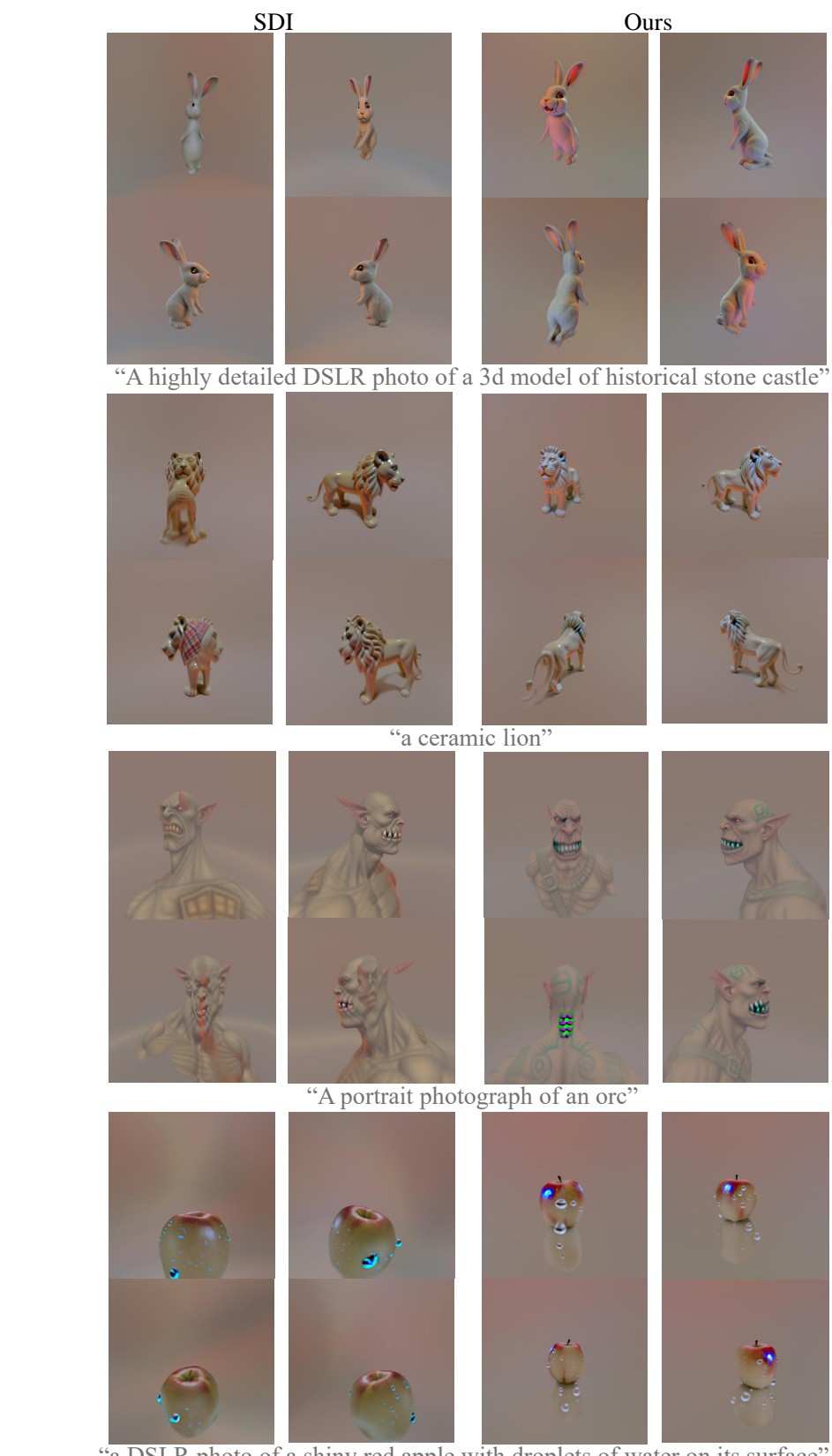

"A highly detailed DSLR photo of a 3d model of historical stone castle"

"a ceramic lion"

"A portrait photograph of an orc"

"a DSLR photo of a shiny red apple with droplets of water on its surface"

Figure 11: Comparison between SDI and SDI combined with OCD (Ours).

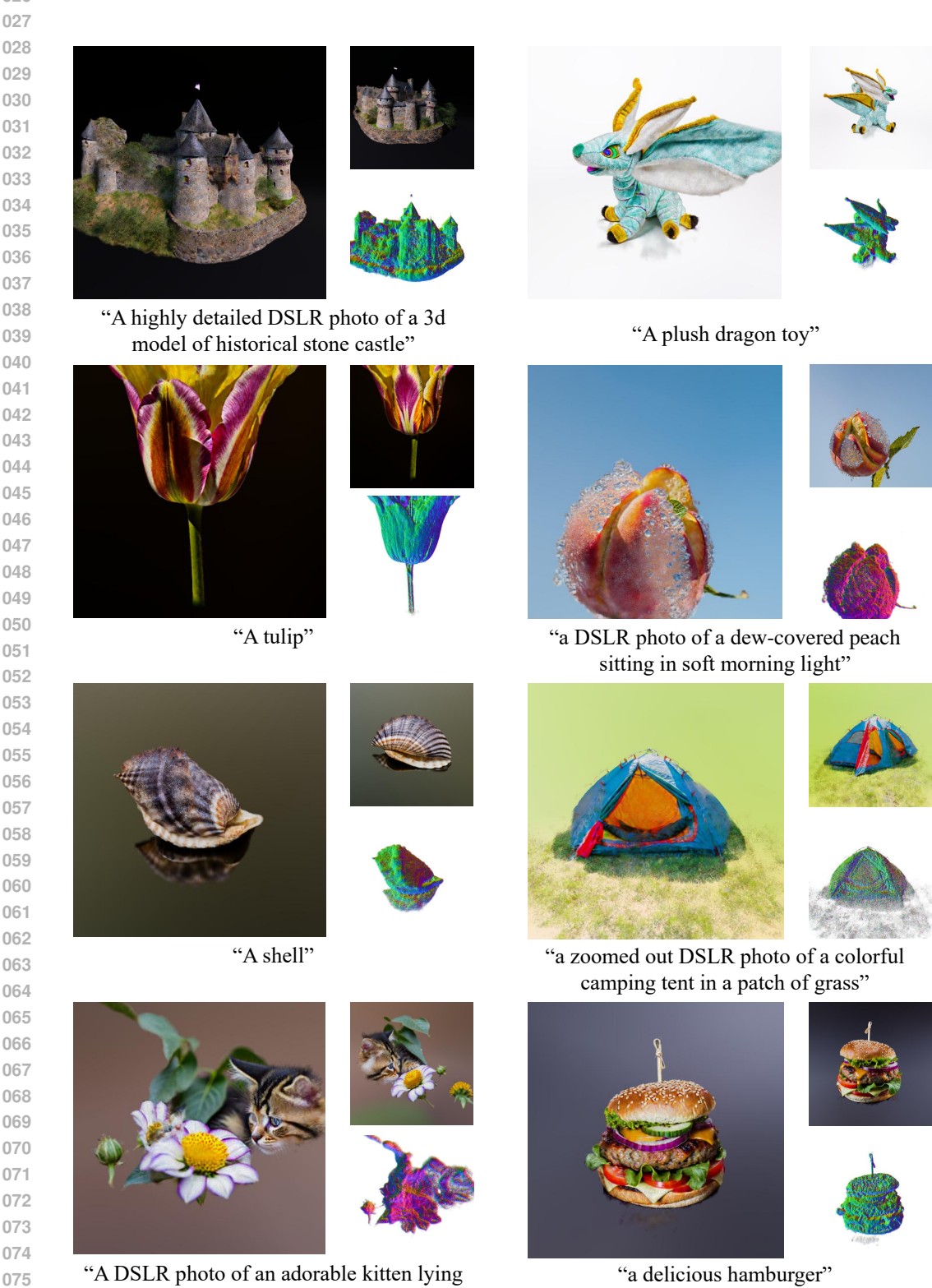

"A highly detailed DSLR photo of a 3d model of historical stone castle"

"A plush dragon toy"

"A tulip"

"a DSLR photo of a dew-covered peach sitting in soft morning light"

"A shell"

"a zoomed out DSLR photo of a colorful camping tent in a patch of grass"

"A DSLR photo of an adorable kitten lying next to a flower"

"a delicious hamburger"

Figure 12: Additional generation results produced by OCD.

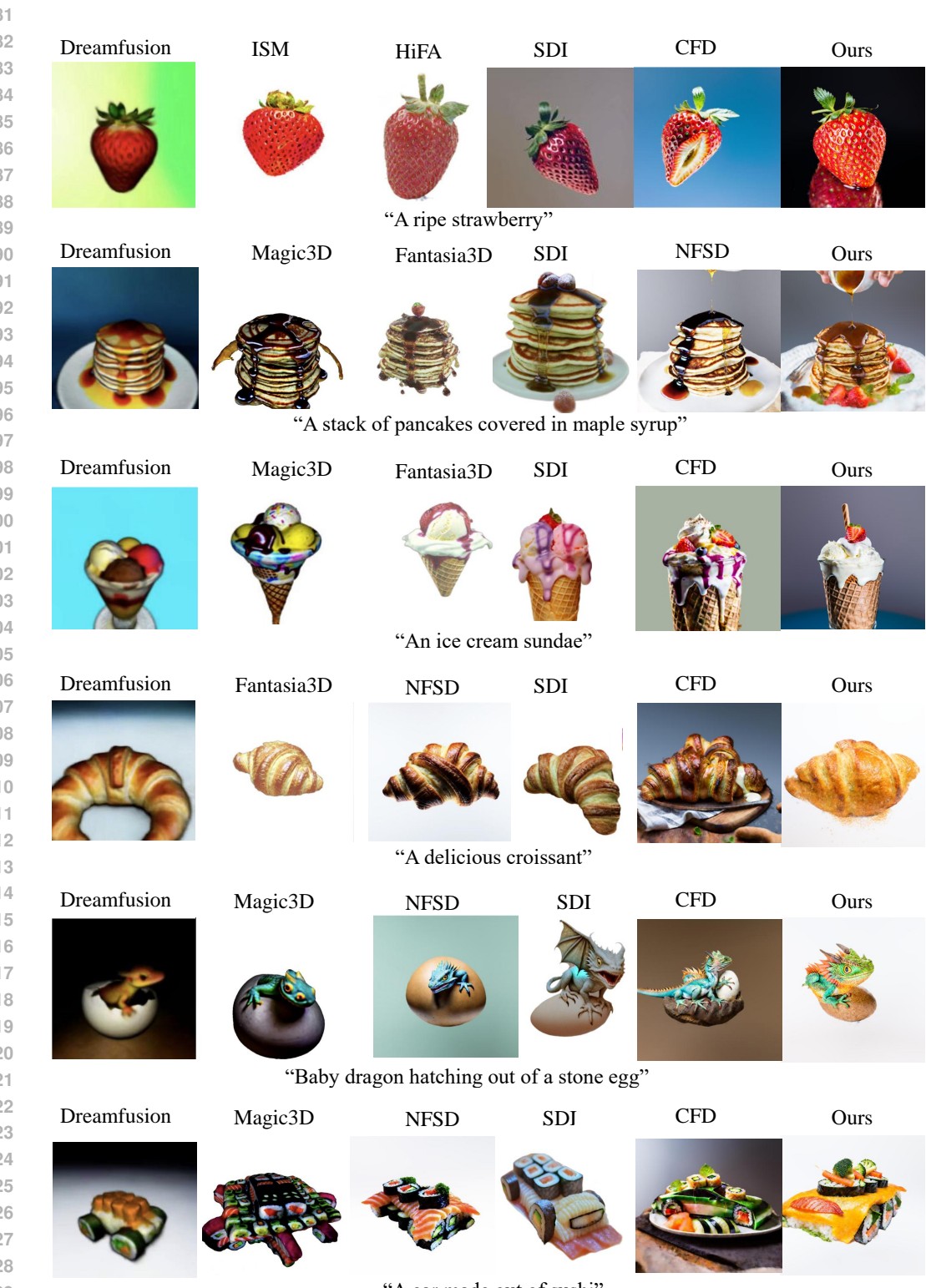

Figure 13: Comparison between OCD and other baselines. Our method employs Algorithm 2 to generate 3D objects at a resolution of 512×512.

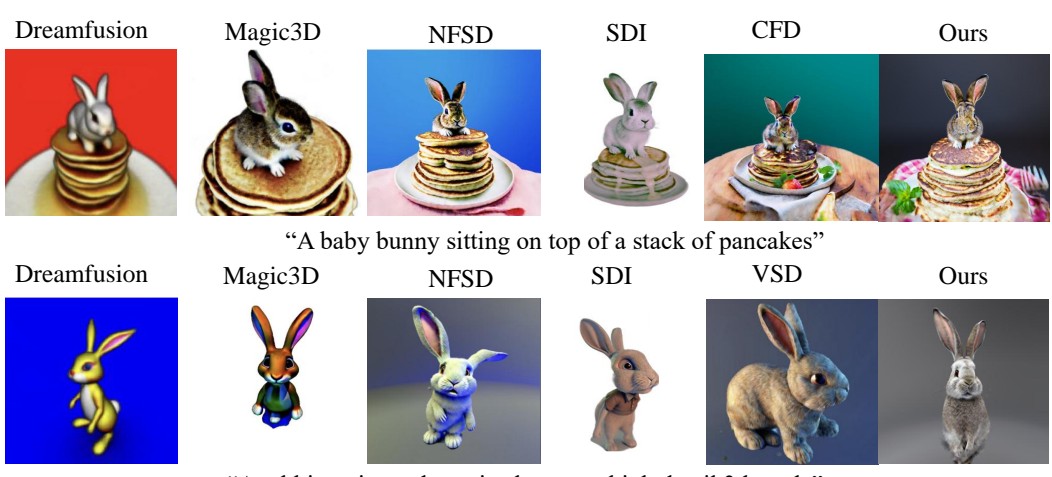

Figure 14: Comparison between OCD and other baselines. Our method employs Algorithm 2 to generate 3D objects at a resolution of 512×512.

