# OpenReview forum: "Object-Consistent Distillation for Text-to-3D Generation"
_ICLR.cc/2026/Conference — ICLR 2026 Conference Withdrawn Submission_

### Official Review · Reviewer_HMsp · 2025-10-31

**Soundness:** 2
**Presentation:** 2
**Contribution:** 2
**Rating:** 4
**Confidence:** 4

**Summary:**

This paper addresses the challenge of object inconsistency in Score Distillation Sampling (SDS) for text-to-3D generation, a problem that often leads to artifacts and distorted models. It proposes Object-Consistent Distillation (OCD), a method that aims to align multi-view pseudo-ground truths with a single 3D object. The method introduces a dynamic object proxy, estimated via a sliding window of recent views. The authors frame their approach with a theoretical model that defines 3D object renderings as a connected subset of image space, which they use to justify their consistency constraint.

**Strengths:**

* The paper clearly and effectively identifies a significant and well-known problem in text-to-3D generation: the object inconsistency across different views that plagues Score Distillation Sampling (SDS).

* The paper provides a new and insightful lens for analyzing the problem by distinguishing between two sources of inconsistency: "cross-view image discrepancy variation" and "cross-view distributional estimation error." By focusing on the former as a previously overlooked issue, the paper offers a novel problem formulation.

**Weaknesses:**

* The paper's experimental validation is critically flawed due to its failure to compare against key state-of-the-art (SOTA) methods that directly address the same problem of geometric consistency, such as MVControl [1], DreamControl [2] and MT3D [3]. This omission makes it impossible to ascertain the true competitive standing or advancement offered by the proposed method.

* The lack of empirical comparison with LucidDreamer further weakens the evaluation. Instead, the paper benchmarks primarily against outdated baselines (e.g., DreamFusion, Fantasia3D) or methods that, while strong in other respects (e.g., HiFA, VSD), are not the most relevant competitors for the specific task of geometric consistency. As a result, outperforming this subset of baselines no longer constitutes a sufficient contribution.

* The central claim of alleviating the Janus problem is not adequately supported by the provided evidence. The paper relies on a few limited qualitative examples, with many of the chosen prompts (e.g., "a delicious croissant," "a hamburger," "a plate of cookies") representing geometrically simple or near-symmetrical objects. These are not challenging or convincing tests for 3D consistency.

* For a paper focused on 3D generation, the absence of more comprehensive evidence, such as 360-degree videos or at least multi-view renders from canonical viewpoints, makes it impossible to fully assess the method's quality and coherence on complex, asymmetric objects.

[1] Li, Zhiqi, et al. "Controllable text-to-3D generation via surface-aligned Gaussian splatting." 2025 International Conference on 3D Vision (3DV). IEEE, 2025.

[2] Huang, Tianyu, et al. "Dreamcontrol: Control-based text-to-3d generation with 3d self-prior." Proceedings of the IEEE/CVF conference on computer vision and pattern recognition. 2024.

[3] Nath, Utkarsh, et al. "Deep Geometric Moments Promote Shape Consistency in Text-to-3D Generation." 2025 IEEE/CVF Winter Conference on Applications of Computer Vision (WACV). IEEE, 2025.

**Questions:**

Please refer to weaknesses

---

### Official Review · Reviewer_QwNk · 2025-11-01

**Soundness:** 2
**Presentation:** 2
**Contribution:** 1
**Rating:** 2
**Confidence:** 4

**Summary:**

This paper addresses the object inconsistency problem in Score Distillation Sampling (SDS) for 3D generation, where pseudo ground truths from different viewpoints fail to correspond to the same 3D object. The authors theoretically model renderings under continuous viewpoints as a connected subset of image space, identify cross-view image discrepancy variation and distributional estimation error as key inconsistency sources, and propose Object-Consistent Distillation (OCD) which enforces consistency by estimating a dynamic object proxy using a sliding window and aligning each viewpoint's rendering toward this proxy, significantly reducing irregular structures and artifacts in generated 3D objects.

**Strengths:**

- The paper is somewhat well-written and easy to follow.
- The experimental (qualitative) results show some effectiveness in conjunction with existing SDS-related methods; however, the scenes shown are quite limited in scope, and many of them are not effective (scenes where front / back / side orientations are unclear, such as cookies or ice cream) in showing improvement in the Janus problem, which this paper attempts to solve.

**Weaknesses:**

- The author correctly notes that the difference in estimation can be strongly attributed to cross-view distributional estimation caused by the inherent randomness of the latent noise given. However, I believe this method does not account for this difference when calculating the means of pseudo-GTs as a proxy direction to which the optimization must proceed. When each viewpoint of the pseudo-GT has randomly sampled noises, the difference induced by randomness in distributional estimation will cause each sampled pseudo-GTs to point to different objects, preventing the optimization towards the mean of sampled pseudo-GTs from having a positive effect in the optimization process. Please further elaborate on this issue.
- The "mean" of pseudo-GT in Section 5 is not clearly defined. Please describe the details by which you acquire the mean for the preceding pseudo-GTs: do you calculate the mean of the latent features within LDMs, or the mean of the RGB images? Either way, I believe that there is a design flaw in this that 1) first, the simple mean of the 2D latent (or image) features does not directly lead to the semantic average/coherence of the sampled viewpoints and 2) second, the mean will quickly fall out of the manifold within the diffusion model in cases where viewpoint differences between pseudo-GT's viewpoints becomes slightly larger, possibly leading to more degradation in the optimization process. Can you show a theoretical proof that the 'mean' of the sampled GTs falls in the ideal manifold within the LDM's distribution so that this method is ensured to have a net positive effect on the optimization process?
- It seems that each optimization step requires a complete denoising process for each sampled pseudo-GT viewpoint (all to train a single viewpoint), which I believe may cause a significant increase in training time and computation cost. Could the authors elaborate on this point, please?
- Lack of quantitative analysis. Please provide some quantitative analysis. The qualitative results are insufficient, as I have briefly mentioned in the Strengths section.
- Typo in L306: Equatio 9 -> Equation 9

**Questions:**

Please see the Weaknesses section.

---

### Official Review · Reviewer_eeqE · 2025-11-03

**Soundness:** 2
**Presentation:** 2
**Contribution:** 2
**Rating:** 2
**Confidence:** 3

**Summary:**

This paper addresses multi-view inconsistency in Score Distillation Sampling (SDS) for text-to-3D generation. The authors propose Object-Consistent Distillation (OCD), which uses a sliding window to average past pseudo-GTs as an "object proxy" and moves current renderings toward this proxy before feeding to the diffusion model. Experiments show improvements over SDS, SDI, and CFD baselines on geometric consistency metrics with claimed <1% overhead.

**Strengths:**

- The method claims <1% additional cost, making it practically attractive.
- Figures in the paper effectively illustrate the multi-view inconsistency problem and provide an intuitive visualization of how 3D objects relate to 2D renderings in image space, making the theoretical framework accessible.

**Weaknesses:**

There is a logical contradiction throughout the paper. The paper explicitly states that pseudo-GTs from different viewpoints correspond to m different 3D objects. Specifically, the gradient formulation shows Eq.8. This clearly identifies that pseudo-GTs x̂₀(c₁), x̂₀(c₂), …, x̂₀(cₙ) belong to different object sets M₁, M₂, …, Mₘ. The paper proposes to create an "object proxy" by averaging these inconsistent pseudo-GTs in Eq.14. If x̂₀(c₁) ∈ M₁ and x̂₀(c₂) ∈ M₂ where M₁ and M₂ represent different 3D objects, why would their average (x̂₀(c₁) + x̂₀(c₂))/2 represent a single consistent object? This is analogous to claiming that averaging a photo of a cat and a photo of a dog produces a coherent representation of a single animal. The paper provides no theoretical justification or empirical evidence for why this averaging operation should yield a consistent object representation rather than simply a blurred, unrealistic mixture. This logical gap undermines the entire methodology. The paper diagnoses that the problem is inconsistency across multiple objects, yet proposes to solve it by averaging those inconsistent representations without explaining why this averaging preserves or creates consistency.

**Questions:**

- The paper explicitly state that pseudo-GTs from different viewpoints correspond to m different 3D objects (Mtar in Eq.8). Your solution averages these pseudo-GTs to create an object proxy (Eq.14). Please provide mathematical proof or empirical evidence for why averaging representations of different objects produces a representation of a single consistent object rather than an unrealistic mixture.
- Alternative explanations not ruled out. The paper doesn't compare against: Simple Gaussian blur (Moving average (Eq. 15) essentially blurs the image by mixing with past pseudo-GTs. Maybe the improvement just comes from removing high-frequency noise, not from the "object consistency" mechanism?), Exponential Moving Average (This is simpler and more standard than sliding window. Why not test it?), Median instead of Mean (If some pseudo-GTs are very wrong (outliers), median would be more robust than mean)

---

### Note · Authors · 2025-12-08

I have read and agree with the venue's withdrawal policy on behalf of myself and my co-authors.